# AUTOREGRESSIVE MODELS RIVAL DIFFUSION MODELS AT ANY-ORDER GENERATION

**Tianqi Du**[1]    **Lizhe Fang**[1]    **Weijie Yang**[2]    **Chenheng Zhang**[1]
**Zeming Wei**[2]    **Yifei Wang**[3]    **Yisen Wang**[1,4]*

[1] State Key Lab of General Artificial Intelligence,
  School of Intelligence Science and Technology, Peking University
[2] School of Mathematical Sciences, Peking University
[3] Amazon AGI SF Lab[†]
[4] Institute for Artificial Intelligence, Peking University

## ABSTRACT

Diffusion language models enable any-order generation and bidirectional conditioning, offering appealing flexibility for tasks such as infilling, rewriting, and self-correction. However, their formulation—predicting one part of a sequence from another within a single-step dependency—limits modeling depth and often yields lower sample quality and stability than autoregressive (AR) models. To address this, we revisit autoregressive modeling as a foundation and reformulate diffusion-style training into a structured multi-group prediction process. We propose Any-order Any-subset Autoregressive modeling (A3), a generalized framework that extends the standard AR factorization to arbitrary token groups and generation orders. A3 preserves the probabilistic rigor and multi-layer dependency modeling of AR while inheriting diffusion models' flexibility for parallel and bidirectional generation. We implement A3 through a two-stream attention architecture and a progressive adaptation strategy that transitions pretrained AR models toward any-order prediction. Experiments on question answering, commonsense reasoning, and story infilling demonstrate that A3 outperforms diffusion-based models while maintaining flexible decoding. This work offers a unified approach for a flexible, efficient, and novel language modeling paradigm. Code is at https://github.com/PKU-ML/Any-order-Any-subset-AR.

## 1 INTRODUCTION

Diffusion language models have recently emerged as a powerful alternative to autoregressive (AR) modeling for text generation (Li et al., 2022; Gong et al., 2025; Gulrajani & Hashimoto, 2023; Sahoo et al., 2024). By iteratively denoising partially masked sequences, they allow any-order generation and bidirectional context conditioning, enabling novel capabilities such as global rewriting, infilling, and self-correction. These properties make diffusion models conceptually appealing for flexible text generation beyond the rigid left-to-right paradigm.

However, diffusion modeling also introduces fundamental limitations. In each denoising step, the model predicts one part of the sequence conditioned on another, forming only a single-layer dependency structure. This simplification weakens the model's ability to capture deep, hierarchical dependencies across tokens. As a result, despite their flexibility, diffusion-based models often underperform AR models in generation quality and training stability (Kim et al., 2025; Sahoo et al., 2024; Zheng et al., 2024). These challenges highlight an inherent trade-off between the expressive power of AR factorization and the flexibility of diffusion-style generation.

Motivated by this observation, we revisit autoregressive modeling as a foundation for constructing a more expressive and stable framework. AR models naturally capture rich, recursive dependencies through sequential conditioning but suffer from their strict left-to-right constraint. We therefore

---

*Corresponding Author: Yisen Wang (yisen.wang@pku.edu.cn).
[†]This work was completed at MIT prior to Yifei Wang joining Amazon.

reformulate diffusion training in a more structured way by extending its two-group prediction to multiple groups, so that the model can preserve AR's multi-layer dependency structure while supporting arbitrary generation orders. This leads to Any-order Any-subset Autoregressive modeling (A3), a unified framework that generalizes standard AR factorization to allow token groups of arbitrary composition and order. A3 retains the probabilistic rigor and stability of AR training while inheriting diffusion models' flexibility for parallel generation and bidirectional context use.

To realize A3 in practice, we design a two-stream attention architecture that supports arbitrary group orderings, and develop a progressive training strategy that adapts pretrained AR models to any-order prediction. It naturally supports diverse inference strategies, including groupwise AR sampling and dynamic resampling, offering a tunable trade-off between generation speed and quality.

Through comprehensive experiments on question answering (Joshi et al., 2017), commonsense reasoning (Zellers et al., 2019; Sakaguchi et al., 2020; Sap et al., 2019; Bisk et al., 2020), and story infilling tasks (Mostafazadeh et al., 2016), we show that A3 achieves strong performance across diverse benchmarks while enabling flexible and efficient generation. Notably, A3 outperforms state-of-the-art diffusion-based models despite using substantially less training data, and demonstrates promising scaling behavior with model size. These results suggest that A3 provides a new direction for bridging the gap between AR and parallel generation paradigms.

Our contributions can be summarized as follows:

- **Conceptually**, we propose A3, a new sequence modeling paradigm that reformulates diffusion-style group prediction into a generalized autoregressive framework, preserving dependency depth while enabling *any-order*, *any-subset* generation.
- **Algorithmically**, we design a two-stream attention architecture and a progressive training recipe that extend pretrained AR models to support arbitrary group orderings.
- **Practically**, we demonstrate that A3 surpasses diffusion-based approaches, offering a flexible and scalable foundation for next-generation language modeling.

## 2 ANY-ORDER ANY-SUBSET AUTOREGRESSIVE MODELING

### 2.1 FORMULATION

In this section, we will demonstrate the formulation of our proposed Any-order Any-subset Autoregressive modeling (A3). Masked diffusion language models provide a flexible approach to sequence generation by iteratively predicting masked subsets of tokens conditioned on visible context (He et al., 2023; Nie et al., 2025). Concretely, during training, given a sequence $x_{1:N}$, the index set $1, 2, \ldots, N$ is partitioned into two disjoint subsets $G_1$ and $G_2$. The model predicts the masked tokens $x_{G_2}$ conditioned on the visible tokens $x_{G_1}$:

$$P(x_{G_2} \mid x_{G_1}) = \prod_{t \in G_2} P(x_t \mid x_{G_1}). \tag{1}$$

This one-step conditional formulation enables parallel prediction and bidirectional context use, supporting any-order generation through repeated resampling of subsets. However, it models only **shallow dependencies**—each token in $G_2$ depends directly on $x_{G_1}$ but not recursively on other predicted tokens. As a result, the global dependency structure is limited to one layer of conditioning, unlike the multi-step compositional nature of autoregressive (AR) factorization.

To enrich the dependency structure while retaining the flexibility of arbitrary grouping, we reformulate this process through an autoregressive lens. Starting from the AR formulation,

$$P(x_{1:N}) = \prod_{t=1}^{N} P(x_t \mid x_{<t}), \tag{2}$$

we generalize the token-wise sequence to a group-wise factorization. The index set $\{1, \ldots, N\}$ is partitioned into $K$ disjoint groups $\{G_1, G_2, \ldots, G_K\}$, and the joint probability is expressed as

$$P(x_{1:N}) = \prod_{k=1}^{K} P(x_{G_k} \mid x_{G_{<k}}). \tag{3}$$

Each group $G_k$ may contain one or more tokens, and the group order is arbitrary. This generalization retains the recursive dependency structure of AR while introducing diffusion-style flexibility. Each factor can be viewed as predicting one subset of tokens from previously generated subsets, but now organized into multiple layers of dependency.

Training the model on randomly sampled group partitions and permutations exposes it to diverse factorization orders and improves robustness to different conditional structures. This resembles the permutation LM objective of XLNet (Yang et al., 2019) but at the group level, enabling richer structural modeling. This formulation bridges diffusion's parallel prediction and AR's hierarchical modeling, forming the foundation of **Any-order Any-subset Autoregressive modeling (A3)**.

## 2.2 DISCUSSION WITH PREVIOUS PARADIGMS

**Comparison with Masked Diffusion Language Models.** Masked diffusion language models (MDLMs) have recently emerged as an alternative to AR decoding, aiming to overcome the sequential bottleneck of one-token-at-a-time generation. By iteratively denoising partially masked sequences, MDLMs can update multiple tokens in parallel and exploit bidirectional context (Austin et al., 2021; Li et al., 2022; Gong et al., 2025; Nie et al., 2025). This flexibility enables controllable, any-order generation and supports tasks such as infilling and global rewriting. However, diffusion approaches face two major limitations. First, inference speed is constrained by the need for many iterative refinement steps, with generation efficiency depending critically on noise schedules and step counts (Gong et al., 2025). While large-scale efforts such as DiffuGPT and LLaDA demonstrate that diffusion LMs can match or surpass AR models in quality, they still require careful tuning and incur nontrivial decoding costs (Gong et al., 2025; Nie et al., 2025). Second, training stability is less favorable than AR: discrete diffusion objectives require complex forward–reverse processes and additional pretraining or adaptation, making optimization more resource-intensive and sensitive to hyperparameters (He et al., 2023).

In contrast, A3 preserves the probabilistic rigor and training simplicity of AR modeling while introducing flexible groupwise factorization. This allows parallel prediction of token subsets without relying on multi-step denoising schedules, thereby offering both efficiency and stability.

**Comparison with AR Multi-Token Prediction.** Another line of work seeks to improve AR efficiency by enabling the model to predict multiple future tokens per step (Gloeckle et al., 2024; Kou et al., 2024; Zhang et al., 2024). Multi-token objectives and speculative decoding significantly reduce inference latency: for instance, predicting four tokens at once can yield up to threefold speedups while preserving or even improving generation quality, particularly in reasoning and code generation tasks (Gloeckle et al., 2024). These methods retain the training stability of standard AR, since the additional objectives can be implemented as auxiliary losses with negligible computational overhead. However, multi-token prediction remains bound to a fixed left-to-right ordering, limiting its modeling flexibility. The approach accelerates sequential decoding but does not enable infilling, bidirectional conditioning, or arbitrary ordering of token generation.

A3 generalizes beyond this paradigm by relaxing the strict AR factorization. Through arbitrary group partitions and orderings, A3 supports both sequential and parallel decoding strategies, combining the efficiency gains of multi-token prediction with greater structural flexibility. For more discussions with related work, refer to Appendix B.

## 3 IMPLEMENTATIONS OF TRAINING AND FLEXIBLE INFERENCE

In this section, we describe the implementation of A3. We begin with the architectural design of A3, where we use a two-stream attention mechanism to enable predictions in arbitrary orders. Next, we present our efficient continuous pretraining strategy, which progressively adapts the model from standard AR prediction to group-based prediction. Finally, we introduce the flexible inference strategy of A3, which leverages its general formulation to support diverse decoding modes.

### 3.1 ARCHITECTURE DESIGN WITH TWO-STREAM ATTENTION

**Limitations of Current Architecture.** The decoder-only Transformer has become the backbone of modern large language models due to its simplicity and effectiveness in next-token prediction.

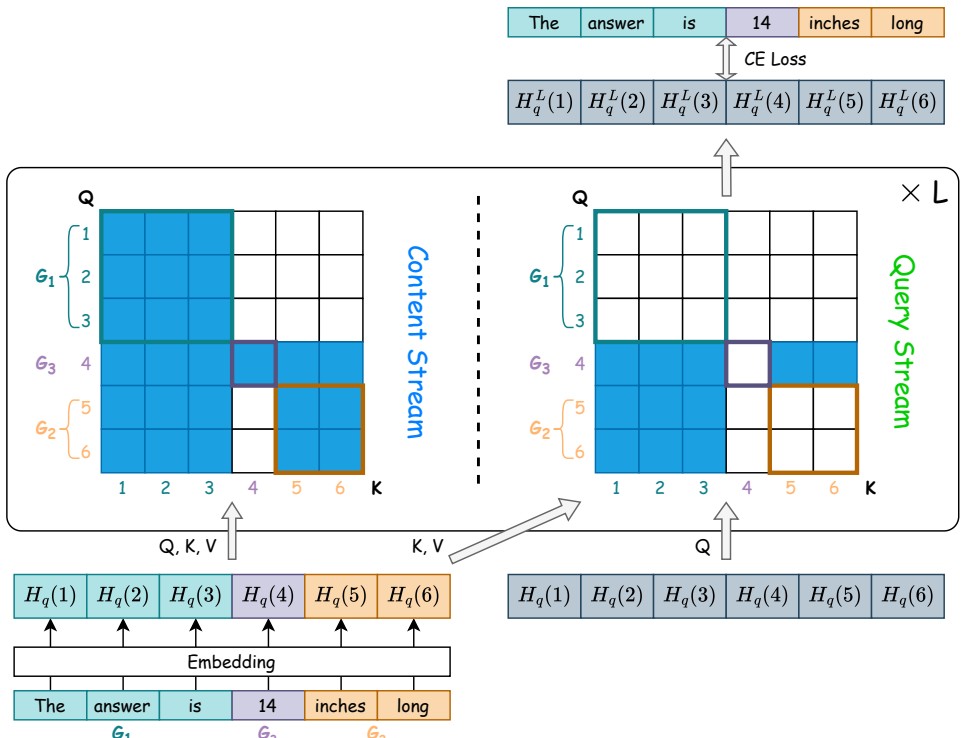

Figure 1: Architecture of the A3 model. Blue entries in the attention mask denote $0$, and white entries denote $-\infty$. The model employs a two-stream attention module with distinct causal masks. The content stream encodes contextual information and attends to tokens within its own group as well as all preceding groups. The query stream encodes positional conditions and attends only to tokens in preceding groups. The final cross-entropy loss is computed between the input context and the query stream's output. For illustration, we provide an example grouping with $G_1 = \{1,2,3\}, G_2 = \{5,6\}, G_3 = \{4\}$, showing how the forward process and causal masks are applied.

It operates with a single stream of hidden states, where information flow is regulated by a causal attention mask. This mask ensures that the representation of the $k$-th token can only attend to the first $k$ tokens, thereby enforcing the AR constraint required for language modeling. While well-suited for the standard left-to-right objective, this design assumes a fixed generation order: given the first $k$ tokens, the model is trained to treat the $(k+1)$-th position as the unique next target. Such rigidity makes it incompatible with any-order prediction, where the next position to be generated need not follow the sequential index.

The encoder-only Transformer, widely used in masked language modeling, represents the opposite design. Rather than causal masking, it processes the full sequence bidirectionally, with missing information represented by mask tokens. Through position embeddings on these masks, the model identifies which locations are to be predicted, allowing arbitrary subsets of tokens to be reconstructed simultaneously. However, this formulation limits dependency modeling: masked positions are predicted in parallel and conditioned only on observed context in a single pass. Without recursive, multi-layered dependencies across tokens, the encoder-only approach struggles to match the generative fidelity of AR models.

**Two-stream Attention Design for Any-order Prediction.** To combine the flexibility of encoder-style masking with the dependency modeling strength of autoregression, A3 extend the two-stream attention mechanism proposed by XLNet (Yang et al., 2019). The model maintains two parallel representations for each position: a **content stream**, which encodes semantic and contextual information from the observed tokens, and a **query stream**, which provides position-aware signals to drive prediction of the next group. This separation allows A3 to retain the recursive structure of autoregression while relaxing the generation order constraint of decoder-only models. Figure 1 illustrates the pipeline of the forward process.

Figure 2: Causal masks for content stream and query stream in different stages. Blue for $0$ and white for $-\infty$. Stage 1: **AR initialization** to reproduce AR factorization. Stage 2: **Group expansion** by allowing groups of size greater than one. Stage 3: **Order permutation** with introducing any-order prediction.

Formally, let $X = (x_1, \ldots, x_N)$ denote the sequence, partitioned into groups $\{G_1, \ldots, G_K\}$. In the **content stream**, the input consists of the observed tokens, embedded and passed through Transformer layers with a designed causal mask. This mask ensures that a token at group $k$ can attend to all tokens in groups $\leq k$, i.e., both its own group and all groups before it. Thus, the content stream at group $k$ aggregates all contextual evidence available up to that point. For group $G_k$, the hidden states in the content stream at layer $l$ are computed as:

$$H_c^{(l)}(i) = \text{Attn}\Big(Q = H_c^{(l-1)}(i), \ K = H_c^{(l-1)}(\leq G_k), \ V = H_c^{(l-1)}(\leq G_k)\Big). \tag{4}$$

In the **query stream**, the input is a shared learnable query vector injected at every position. The key and value matrices are tied to those of the content stream, while the queries are separate. With an appropriately designed causal mask, each query vector at group $k$ can only attend to content tokens in groups $< k$, not including its own group. This forces the query representation to serve as the position-aware predictor for the tokens in group $k$, relying only on prior context rather than future information. Conceptually, the query stream specifies *where* to predict (positional conditioning), while the content stream provides *what* to predict (contextual grounding). The hidden states in the query stream at layer $l$ for group $G_k$ are:

$$H_q^{(l)}(i) = \text{Attn}\Big(Q = H_q^{(l-1)}(i), \ K = H_c^{(l-1)}(< G_k), \ V = H_c^{(l-1)}(< G_k)\Big), \tag{5}$$

where the initialization $H_q^{(l-1)}(i) = w$ is a learnable query vector shared across positions, and the causal mask ensures the query stream at group $k$ can only access content states from strictly earlier groups.

Finally, the predictive distribution for token $x_i \in G_k$ is parameterized by:

$$p(x_i \mid X_{<G_k}) = \text{Softmax}\Big(W \cdot H_q^{(L)}(i)\Big), \tag{6}$$

where $L$ is the final layer and $W$ projects the query hidden state to the vocabulary.

### 3.2 MULTI-STAGE TRAINING WITH PROGRESSIVE TOKEN GROUPING

Building on the connection between standard AR and A3, we design a progressive adaptation strategy that smoothly transitions from left-to-right generation to fully flexible any-order prediction. To leverage the stability and strong initialization of existing AR models, we begin training A3 from a pretrained AR checkpoint and gradually relax its constraints through three stages:

- **Stage 1: AR Initialization.** We align A3 with conventional AR training by setting the two-stream causal masks to exactly reproduce left-to-right factorization (Figure 2 Stage 1). Formally, the sequence $x_{1:N}$ is partitioned into singleton groups:

$$G_1 = \{1\}, \ G_2 = \{2\}, \ \ldots, \ G_N = \{N\}. \tag{7}$$

  This ensures that $P(x_{1:N}) = \prod_{t=1}^{N} P(x_t \mid x_{<t})$, identical to standard AR, providing a stable initialization.

- **Stage 2: Group Expansion.** We expand beyond token-level prediction by allowing groups of size greater than one (Figure 2 Stage 2). Concretely, the sequence is partitioned into contiguous segments of fixed size $s > 1$, e.g.,

$$G_1 = \{1, \ldots, s\}, \quad G_2 = \{s+1, \ldots, 2s\}, \quad \ldots \tag{8}$$

  with $s$ gradually increased from 1 to 4. This teaches the model to predict multiple tokens jointly within each group while still maintaining AR dependencies across groups.

- **Stage 3: Order Permutation.** We introduce any-order prediction within groups (Figure 2 Stage 3). The group structure $G_1, G_2, \ldots, G_K$ remains sequential, but the token indices assigned to each group are drawn from a random permutation of $\{1, \ldots, N\}$. For example, if $\pi$ is a random permutation of indices, then

$$G_1 = \{\pi(1), \ldots, \pi(s)\}, \quad G_2 = \{\pi(s+1), \ldots, \pi(2s)\}, \quad \ldots \tag{9}$$

  The model therefore learns to predict tokens in arbitrary subsets, while still preserving a group-to-group AR factorization:

$$P(x_{1:N}) = \prod_{k=1}^{K} P(x_{G_k} \mid x_{G_{<k}}). \tag{10}$$

  This exposes the model to diverse intra-group orderings and enables it to generalize to arbitrary prediction targets at inference.

By the end of this curriculum, the model is able to predict arbitrary subsets of tokens as coherent groups while preserving the recursive dependency structure of AR. Importantly, at every stage of training, each token in the sequence belongs to exactly one group, so all tokens are always predicted, maximizing the learning signal and computational efficiency.

## 3.3 FLEXIBLE INFERENCE VIA GROUPWISE DECODING

Building on the A3 formulation, we propose flexible inference strategies that extend beyond conventional AR decoding. The first decoding method we introduce is **groupwise AR sampling**, which generalizes standard left-to-right generation by sampling groups of tokens sequentially rather than strictly one-by-one. Formally, let the token positions of a sequence be partitioned into $K$ groups $\mathcal{G} = \{G_1, G_2, \ldots, G_K\}$, where $G_k \subseteq \{1, \ldots, n\}$ and $\bigcup_{k=1}^{K} G_k = \{1, \ldots, n\}$. Given a prompt covering groups $G_1, \ldots, G_{k_0}$, the model generates subsequent groups by conditioning on all preceding groups:

$$p_\theta(x_{G_{k_0+1}}, \ldots, x_{G_K} \mid x_{G_{\leq k_0}}) = \prod_{k=k_0+1}^{K} p_\theta(x_{G_k} \mid x_{G_{<k}}). \tag{11}$$

Here, $x_{G_k}$ denotes the tokens within group $G_k$, and $x_{G_{<k}}$ the tokens of all earlier groups. This reduces to the classical AR factorization when $|G_k| = 1$ for all $k$, but naturally generalizes to larger groups. The procedure is summarized in Algorithm 1. Concretely, several grouping strategies can be applied:

1. **Token-wise grouping.** Each token is treated as its own group, i.e., $G_k = \{k\}$. The decoding reduces to the standard left-to-right AR generation:

$$p_\theta(x_1, \ldots, x_n) = \prod_{t=1}^{n} p_\theta(x_t \mid x_{<t}). \tag{12}$$

2. **Fixed-size grouping.** Tokens are partitioned into groups of size $s$, e.g., $G_k = \{(k-1)s + 1, \ldots, ks\}$ for $s \in \{2, 4\}$. In this case, the model predicts $s$ tokens jointly per step and accelerates decoding.

3. **Task-specific grouping.** For infilling tasks we allow groups to be arbitrary index subsets and then assign group ids so that groups containing masked positions are decoded after groups used as context. Concretely, let the sequence be partitioned into left, middle and right index sets $L, M, R$ (so $\{1, \ldots, n\} = L \cup M \cup R$). We choose an index $k_0$ such that

| **Algorithm 1** Groupwise AR Sampling | **Algorithm 2** Dynamic Resampling |
|---|---|
| **Require:** Prompt tokens $x_{1:m}$, grouping strategy $\mathcal{G} = \{G_1, G_2, \ldots, G_K\}$, model $f_\theta$ | **Require:** : Prompt tokens, model $f_\theta$, criterion |
| **Ensure:** Generated sequence $\hat{x}_{1:n}$ | 1: Initialize $F_0$ with prompt tokens, $U_0$ with blank positions |
| 1: Initialize $\hat{x}_{1:m} \leftarrow x_{1:m}$ | 2: **while** $U_t \neq \emptyset$ **do** |
| 2: Find the last group index $k_0$ in the prompt | 3:    **for** each $i \in U_t$ **do** |
| 3: **for** $k = k_0 + 1$ to $K$ **do** | 4:        Compute $p_\theta(x_i \mid x_{F_t})$ |
| 4:    Compute context representation $h \leftarrow f_\theta(\hat{x}_{G_{<k}})$ | 5:    **end for** |
| 5:    Sample tokens $\hat{x}_{G_k} \sim p_\theta(\cdot \mid h)$ | 6:    Select subset $S_t \subseteq U_t$ based on criterion |
| 6: **end for** | 7:    **for** each $i \in S_t$ **do** |
| 7: **return** Completed sequence $\hat{x}_{1:n}$ | 8:        Sample $\hat{x}_i \sim p_\theta(x_i \mid x_{F_t})$ |
|  | 9:    **end for** |
|  | 10:    Update $F_{t+1} \leftarrow F_t \cup S_t, U_{t+1} \leftarrow U_t \setminus S_t$ |
|  | 11: **end while** |
|  | 12: **return** Completed sequence $x_{1:n}$ |

every group $G_k$ satisfying $G_k \cap M = \emptyset$ has $k \leq k_0$, while every group that contains any masked position satisfies $k > k_0$ (groups need not be contiguous and a single group may contain tokens from both $L$ and $R$). Under this design, all context groups (those covering $L$ and $R$) appear before the masked groups, and the model performs:

$$p_\theta(x_M \mid x_{G_{\leq k_0}}) = \prod_{k=k_0+1}^{K} p_\theta(x_{G_k} \mid x_{G_{<k}}), \tag{13}$$

which realizes AR dependencies inside each masked group while conditioning on both left and right contexts. This flexible assignment enables infilling where context groups are formed from arbitrary subsets of $L \cup R$, and masked spans are predicted group-by-group. This capability distinguishes A3 from conventional AR models, which cannot directly condition on future context during generation.

**Dynamic Resampling Inference.** Beyond fixed grouping, A3 also supports a more adaptive inference procedure inspired by iterative refinement (Li et al., 2022; Chen et al., 2024a). Here, the grouping $\mathcal{G}$ is not fixed. At each step, the model evaluates all unfinished positions simultaneously, conditioned on the completed tokens. Formally, suppose $U_t \subseteq \{1, \ldots, n\}$ is the set of unfinished (blank) positions at iteration $t$, and $F_t$ is its complement of finished positions. The model computes predictive distributions

$$p_\theta(x_i \mid x_{F_t}), \quad \forall i \in U_t. \tag{14}$$

Based on these distributions, we then select a subset $S_t \subseteq U_t$ to be committed at this step, according to some criterion such as maximum confidence, lowest entropy (Kim et al., 2025), or simply random sampling. Once $S_t$ is chosen, the tokens at $S_t$ are sampled and added to the finished set:

$$F_{t+1} = F_t \cup S_t, \quad U_{t+1} = U_t \setminus S_t. \tag{15}$$

This process repeats until $U_T = \emptyset$, at which point the sequence is fully generated. The procedure is summarized in Algorithm 2. The advantage of this dynamic resampling strategy is twofold. First, it allows the model to adaptively choose the granularity of generation based on prediction confidence, committing to easy tokens early while deferring more uncertain positions until later. Second, unlike diffusion-style denoising which follows a pre-specified noise schedule, A3 inference directly uses the conditional distributions defined by the AR factorization, ensuring consistency between training and inference.

These inference strategies highlight a trade-off between efficiency and flexibility. Fixed-group sampling is fast but less adaptive, as performance depends on group alignment with text structure. Dynamic resampling is slower since all unfinished positions are reevaluated at each step, but it yields greater accuracy by adapting token commitment to model confidence. We will compare these strategies in the next section on real-world tasks.

Table 1: Comprehensive evaluation of different language models. There are 4 types of these models: AR for autoregressive, DD for discrete diffusion, CD for continuous diffusion and A3 for our proposed model. For the infilling task, we use ROUGE-1/2/L score; for other tasks, we use the accuracy (%) metric. * refers to the results reported in DiffuLlama (Gong et al., 2025).

| Model | Size | Type | QA | CommonSense Reasoning | | | | Infilling |
| | | | TriQA | HSwag | Wino. | SIQA | PIQA | ROCStories |
|---|---|---|---|---|---|---|---|---|
| Llama-3.1 | 8B | AR | 52.1 | 76.0 | 63.9 | 46.7 | 80.3 | 11.7/2.3/10.5 |
| Plaid* | 1B | CD | 1.2 | 39.3 | 51.3 | 32.3 | 54.5 | 12.1/1.1/11.2 |
| Dream | 7B | DD | 18.3 | 26.9 | 51.8 | 36.6 | 55.8 | 11.7/2.3/10.5 |
| DiffuLlama* | 7B | DD | 18.5 | **58.7** | 56.4 | 43.2 | 63.3 | **23.3/5.5/21.2** |
| | 1B | A3 | 10.2 | 40.2 | 52.8 | 35.1 | 64.7 | 11.8/1.7/11.1 |
| A3 | 3B | A3 | 15.9 | 49.6 | 54.3 | 38.9 | 70.1 | 11.3/2.3/10.2 |
| | 8B | A3 | **19.4** | 58.4 | **60.2** | **45.2** | **78.1** | 19.2/4.6/18.6 |

Table 2: Conditional generation quality (measured by perplexity using Llama-3.1-8B) of previous diffusion models with A3 using different dynamic sampling strategy.

| Model | Random | | Confidence | | Entropy | |
| | step=512 | step=1024 | step=512 | step=1024 | step=512 | step=1024 |
|---|---|---|---|---|---|---|
| Dream | **58.4** | **46.2** | 21.3 | 17.2 | 18.7 | 16.4 |
| DiffuLlama | 72.3 | 58.4 | 24.1 | 18.3 | 20.9 | 14.3 |
| A3-8B | 66.4 | 49.3 | **20.1** | **16.8** | **14.3** | **11.2** |

## 4 EXPERIMENTS

### 4.1 SETUP

**Training Setup.** We initialize our models from the LLaMA series, including LLaMA-3.1-8B, LLaMA-3.2-3B, and LLaMA-3.2-1B (Grattafiori et al., 2024). For training data, we construct a mixture of the FineWeb dataset (Penedo et al., 2024) and the SlimPajama dataset (Soboleva et al., 2023), following prior work on DLMs and AR models. From this mixture, we sample 8B tokens and apply sequence packing with a maximum context length of 2048. All models are trained with full-parameter fine-tuning in `bf16`. In the progressive adaptation recipe, the first two training stages are trained for one epoch over 20% of the dataset, while the final stage is trained for one epoch over the full dataset. Additional training details are provided in Appendix A.

**Evaluation Setup.** We adopt the evaluation protocol of Gong et al. (2025) to compare our models against both diffusion and AR baselines. For reading comprehension, we evaluate on TriviaQA (Joshi et al., 2017) using exact match accuracy. For commonsense reasoning, we consider HellaSwag (Zellers et al., 2019), Winogrande (Sakaguchi et al., 2020), SIQA (Sap et al., 2019), and PIQA (Bisk et al., 2020), all assessed by multiple-choice accuracy. For story infilling, we use ROCStories (Mostafazadeh et al., 2016) and report ROUGE scores (Lin, 2004). We compare against two categories of baselines: (a) the base AR model LLaMA-3.1-8B, and (b) recent diffusion language models of varying sizes, including Plaid-1B (Gulrajani & Hashimoto, 2023), Dream-7B (Ye et al., 2025), and DiffuLlama-7B (Gong et al., 2025).

### 4.2 MAIN RESULTS

The results in Table 1 show that A3 consistently outperforms diffusion-based models across QA, commonsense reasoning, and infilling tasks. For example, A3-8B achieves 19.4 accuracy on TriviaQA and 78.1 on PIQA, surpassing all the diffusion baselines, while also attaining competitive ROUGE scores for story infilling. These gains are particularly noteworthy given that A3 is trained on only 8B tokens, whereas DiffuLlama is trained on 65B. Although A3 still underperforms the AR baseline, this gap is likely attributable to limited training data; with larger-scale pretraining, we expect A3 to close the difference further.

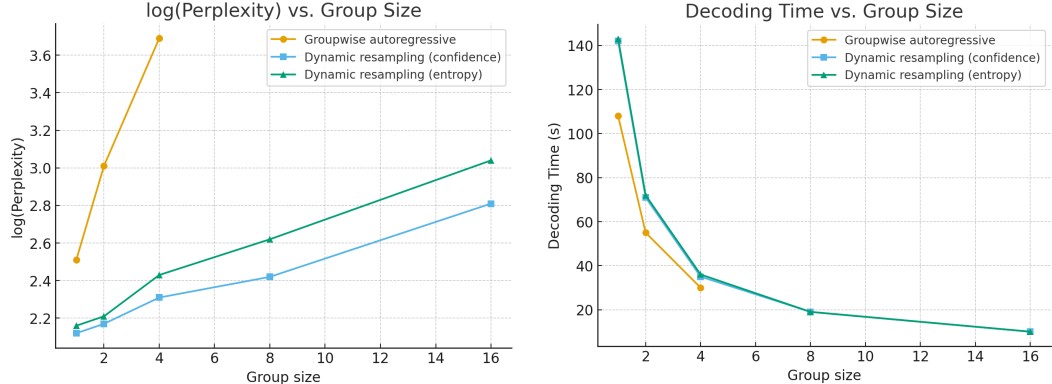

Figure 3: Unconditional generation log(perplexity) and speed using A3-8B. The perplexity is measured by Llama-3.1-8B and we compare several decoding strategies. Dynamic resampling will cost more time but have lower perplexity.

Table 3: Performance with different training curriculum schedule. We evaulate two variants trained on 2B tokens: 1. original curriculum and, 2. skipping stage 1 and 2 (directly training on stage 3: order permutations).

| Schedule | QA | CommonSense Reasoning | | | | Infilling |
|---|---|---|---|---|---|---|
| | TriQA | HSwag | Wino. | SIQA | PIQA | ROCStories |
| Original | **15.6** | **49.3** | **56.7** | **39.6** | **69.4** | **13.2/2.3/12.6** |
| Skipping Stage 1 & 2 | 11.3 | 44.2 | 54.1 | 37.3 | 64.2 | 13.1/2.2/12.4 |

Importantly, A3 demonstrates clear scaling behavior: performance improves steadily from 1B to 3B to 8B parameters, indicating that the method benefits from larger models in the same way as conventional AR training. Overall, these results confirm that A3 offers a favorable balance between AR and diffusion paradigms, combining strong reasoning accuracy with flexible generation, and holds promise for further improvements under larger-scale training.

Additionally, we compare A3-8B with the diffusion models Dream and DiffuLlama on conditional generation quality. For each model, we use the same 100 randomly sampled 128-token prefixes from the Pile (Gao et al., 2020) as conditioning inputs. Each model then generates a 2048-token continuation, and we evaluate the quality of the generated text using perplexity computed by Llama-3.1-8B. The results are summarized in Table 2. While Dream achieves slightly lower perplexity under the random sampling schedule, A3-8B consistently outperforms both baselines when using confidence-or entropy-based dynamic sampling, demonstrating clear gains from our sampling strategy.

## 4.3 ABLATION STUDY

**Sampling Strategies.** To better understand the trade-offs between the two proposed inference strategies in Section 3.3, we conducted unconditional generation experiments under the A3 decoding framework. For groupwise AR sampling, we vary the group size from 1 to 4. For dynamic resampling, we vary the group size from 1 to 16 and implemented two selection criteria: (1) Confidence-based: selecting positions with highest maximum softmax probability. (2) Entropy-based: selecting positions with minimum output entropy. For each sequence, we sample with temperature of 1.5 and top-p of 0.95. Figure 3 reports the log of perplexity measured by Llama-3.1-8B and the average decoding time for one sequence.

We observe that dynamic resampling methods consistently achieve lower perplexity than groupwise AR sampling, indicating that they produce higher-quality generations. The confidence-based and entropy-based criteria yield very similar performance, with confidence being slightly better at smaller group sizes. However, all strategies show a trend of increasing perplexity as group size grows, reflecting the trade-off between decoding granularity and modeling accuracy. We can also

Table 4: Performance of A3 with different training data on TriviaQA and perplexity measured by Llama-3.1-8B.

| Model | TriviaQA | log(Perplexity) |
|---|---|---|
| A3 (6B tokens) | 16.2 | 2.9 |
| A3 (8B tokens) | 19.4 | 2.5 |
| A3 (10B tokens) | 22.5 | 2.3 |
| AR (15T) | 52.1 | 0.8 |

Table 5: Model loss of A3 across context lengths, which is stably small.

| Length | Model loss |
|---|---|
| 256 | 3.54 |
| 512 | 3.51 |
| 1024 | 3.34 |
| 2048 | 3.23 |

see that decoding time decreases sharply with larger group sizes. Groupwise AR sampling is fastest at the same group size because it only generates the designated group per step, while dynamic resampling requires evaluating all unfinished tokens at each iteration, making it slower. However, as group size increases, dynamic resampling speeds up considerably, nearly matching the efficiency of groupwise sampling at large group sizes.

Overall, these results demonstrate a speed–accuracy trade-off. Groupwise AR sampling is faster but less accurate, while dynamic resampling achieves better perplexity at the cost of slower decoding. Importantly, A3 provides the flexibility to choose between these strategies depending on the requirements of the application, making it more flexible than conventional AR or diffusion-based methods.

**Curriculum schedule.** A3 introduces a different causal mask and attention flow from a standard AR transformer, and the model must progressively adapt from strict left-to-right prediction to multi-token and eventually arbitrary-order factorization. To assess the sensitivity of the schedule, we train two variants on 2B tokens: 1. original curriculum, and 2. skipping stage 1 and 2 (directly training on stage 3: order permutations). Results are shown in Table 3. Skipping the early stages consistently hurts performance by 4–6 points on several benchmarks, which proves the importance of such adaptation stage. An adaptive schedule, e.g., based on training loss, may further improve robustness. We plan to investigate this direction in the future work.

**Performance with more data.** Since the training budget for A3 is much less than the baseline (2B for A3, 60B for DiffuLlama and 15T for Llama-3.1-8B), in order to isolate the architecture effect on the worse performance than the AR baseline, we track how A3 improves under increasing post-training data. We use 6B, 8B (default) and 10B tokens to train A3. The results are shown in Table 4. Performance increases steadily with more data. This confirms that A3 benefits strongly from data scale and that the gap to fully-trained AR models is due to training budget, not a limitation of the A3 architecture.

**Robustness on context length.** In order to investigate whether A3 is robust across different context lengths, we input contexts with length of 512, 1024 and 2048 to A3 and calculate the loss. The results are shown in Table 5. The model loss keeps stable within the training length, indicating the robustness of A3 across different context lengths.

## 5 CONCLUSION

We have presented Any-order Any-subset Autoregressive modeling (A3), a novel framework that generalizes traditional autoregressive factorization to enable flexible, groupwise generation of tokens in arbitrary orders. By combining a two-stream attention architecture with a progressive training strategy, A3 achieves the dual goals of generation flexibility and modeling stability. Our approach supports a wide range of decoding strategies, including groupwise autoregressive sampling and dynamic resampling, offering a tunable trade-off between speed and accuracy. Through comprehensive experiments, we demonstrate that A3 outperforms diffusion-based models in reasoning, question answering, and infilling tasks. These results highlight A3's ability to balance efficiency, flexibility, and quality, making it a promising direction for future sequence modeling. In the future, we plan to explore scaling A3 to larger models and datasets, as well as applying it to more challenging tasks such as long-context reasoning.

## ACKNOWLEDGEMENTS

Yisen Wang was supported by National Natural Science Foundation of China (92370129, 62376010), Beijing Major Science and Technology Project under Contract no. Z251100008425006, Beijing Nova Program (20230484344, 20240484642), and State Key Laboratory of General Artificial Intelligence. Yifei Wang was supported in part by the NSF AI Institute TILOS (NSF CCF-2112665), and an Alexander von Humboldt Professorship. Zeming Wei was supported by Beijing Natural Science Foundation (QY24035).

## ETHICS STATEMENT

This work complies with the ICLR Code of Ethics. While our methods are general, they may be applied in contexts with societal implications, including risks related to bias, fairness, and privacy. We encourage responsible use and declare no conflicts of interest.

## REPRODUCIBILITY STATEMENT

We provide detailed descriptions of our methodology, datasets, model configurations, and evaluation metrics in both the main text and the Appendix. Codes will be released upon acceptance.

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

## A TRAINING HYPERPARAMETERS

We list the hyperparameters in the training stage in Table 6

Table 6: The hyperparameter list

| Hyperparameter | Value |
|---|---|
| *Training* | |
| Batch Size | 64 |
| Epoch | [0.2, 0.2, 1] |
| Optimizer | AdamW |
| LR | 2e-5 |
| Betas | (0.9, 0.999) |
| Weight Decay | 0.01 |
| LR Schedule | WarmupLR |
| Warmup Iters | [50, 50, 50] |
| Max Sequence Length | 2048 |
| *Sampling (Section 4.3)* | |
| Top-p | 0.95 |
| Temperature | 1.5 |

## B ADDITIONAL RELATED WORK

**Continue Pre-training.** Pretraining large language models has been proven to be complex and computationally expensive (Samragh et al., 2024). Consequently, continued pre-training has been proposed as an effective method to adapt existing large language models to specific domains (Ke et al., 2023; Gururangan et al., 2020; Guo et al., 2025) or to endow them with new capabilities, such as handling longer contexts (Chen et al., 2024b; Fu et al., 2024; Xiong et al., 2023; Fang et al., 2025) or code generation (Xu et al., 2024). Notably, certain continued pre-training efforts, such as those in scaling diffusion language models (Gong et al., 2025), have transcended autoregressive (AR) language modeling by converting large language models into diffusion-based architectures, thereby enabling parallel token generation. In contrast, our work retains autoregressive language modeling but innovatively incorporates a two-stream architecture and semi-autoregressive decoding to similarly support parallel prediction of multiple tokens, achieving significant reductions in inference latency compared to both autoregressive and diffusion-based baselines.

**Diffusion Language Model.** Diffusion models (Sohl-Dickstein et al., 2015; Ho et al., 2020; Song et al., 2020) have demonstrated remarkable capabilities in image generation. Consequently, a series of works have sought to extend diffusion models to text generation, which can be roughly divided into the continuous diffusion model and the discrete diffusion model. One straightforward approach involves embedding text data into a continuous space and directly applying diffusion models (Li et al., 2022; Gong et al., 2022; Han et al., 2022; Dieleman et al., 2022). However, the scalability of continuous diffusion methods remains a challenge, as they require substantially greater computational cost compared to AR models to achieve equivalent performance(Gulrajani & Hashimoto, 2023). To better accommodate the discrete nature of text, an alternative paradigm replaces continuous diffusion with a discrete process, introducing an absorbing [MASK] state as noise (Austin et al., 2021; Hoogeboom et al., 2021; Zheng et al., 2024; Sahoo et al., 2024). Lou et al. (2023) demonstrated that masked diffusion models (MDMs) achieve perplexity comparable to or even surpassing that of AR models at the GPT-2 scale. Ou et al. (2025) established foundational theoretical results, affirming the feasibility of MDMs. In comparison to MDMs, our method similarly enables parallel prediction of groups of tokens and leverages bidirectional context. However, by retaining autoregressive modeling, our approach utilizes every token during training, thereby facilitating faster convergence relative to MDMs.

**Non-autoregressive Generation.** Non-autoregressive (NAR) generation (Gu et al., 2017) accelerates inference by producing target tokens in parallel, eliminating the dependency on previously generated tokens inherent in traditional AR models. This approach substantially improves genera-

tion speed, but potentially at the cost of some accuracy. Conventional NAR models have focused mainly on machine translation and automatic speech recognition, emphasizing the trade-off between speed and quality (Higuchi et al., 2020; Lee et al., 2018). For instance, Guo et al. (2020) employed sequence-to-sequence labels in translation tasks, training NAR models via a curriculum learning strategy on attention masks. However, early NAR methods in text generation remain constrained by issues of global coherence and diversity. In recent years, insertion-based models(Stern et al., 2019; Gu et al., 2019) and diffusion language models (Gong et al., 2022; Lou et al., 2023; Shen et al., 2023) have progressively addressed these limitations. For example, SSD-LM (Han et al., 2022) leverages a diffusion-based language model to generate text blocks in a semi-autoregressive manner, enabling local bidirectional context updates. Compared to diffusion language models, our method similarly achieves parallel multi-token prediction and bidirectional context modeling, while benefiting from faster training convergence due to the absence of masked tokens during training.

**Multi-token Prediction.** Multi-Token Prediction (MTP) seeks to move beyond strictly stepwise next-token prediction by enabling models to forecast multiple future tokens per autoregressive step, improving inference efficiency with minimal quality degradation. Some methods achieve MTP by modifying the Transformer architecture. For instance, Stern et al. (2018) demonstrated through blockwise decoding that multi-token acceptance is feasible when supported by verification mechanisms. Medusa (Cai et al., 2024) add parallel heads to predict multi-step continuations with lightweight acceptance rules. Other approaches realize MTP via multi-model collaboration. Speculative decoding (Leviathan et al., 2023) uses a draft model to propose multi-token candidates that the main model verifies, while EAGLE-like validators (Li et al., 2024) further strengthen multi-token verification. Our method similarly modifies the Transformer structure, but innovatively leverages a two-stream architecture and semi-autoregressive decoding to achieve MTP, significantly reducing generation latency while attaining higher accuracy than baselines.

## C  THE USE OF LARGE LANGUAGE MODELS (LLMS)

In this work, LLMs are primarily employed for polishing the language of the manuscript to ensure grammatical correctness and coherence. Importantly, all conceptual development, experimental design, and result interpretation are conducted independently by the authors. The use of LLMs is strictly limited to auxiliary tasks, ensuring that the scientific contributions of this paper remain entirely unaffected by such tools.

## D  MORE ANALYSIS EXPERIMENTS

**Comparison with speculative decoding.** We additionally conduct generation experiments as in Figure 3. We use speculative decoding (Leviathan et al., 2023) with Llama-3.2-1B as the draft model and Llama-3.1-8B as the target model, matching A3's group size (max 4). Perplexity is measured by Llama-3.1-70B. Results are shown in Table 7. Speculative decoding achieves lower perplexity but at higher wall-clock cost, while A3 provides competitive quality with faster decoding in this setting.

Importantly, semi-AR methods such as speculative decoding and multi-token prediction are decoding-level accelerators: they maintain the same left-to-right AR factorization and improve efficiency through draft-model proposals or multi-step predictions. In contrast, A3 changes the model factorization itself by enabling groupwise and permutation-based prediction. This makes A3 **orthogonal** to speculative/MTP: these accelerations can also be applied on top of A3's factorization in principle.

**Results on longer contexts.** To evaluate whether A3 remains stable under significantly longer contexts than 2k tokens, we finetuned both Llama-3.1-8B and A3-8B on 8k-length sequences from PG19 (Rae et al., 2019) using the same training budget (100 steps, batch size 64). We then evaluated them on the single-document QA task from LongBench v1 (Bai et al., 2024), which requires reasoning over long passages. We used dynamic sampling for A3 with group size 1 and group size 2. The results are shown in Table 8.

A3 (group size 1) improves over the AR baseline with 1.7%, suggesting that the A3 factorization does not degrade long-context modeling and may offer small gains without parallel decoding. A3

Table 7: Comparison between speculative decoding and A3 on generating perplexity and time.

|  | log(perplexity) | Time |
|---|---|---|
| Speculative decoding | **1.9** | 1.2× |
| A3 | 2.1 | **1**× |

Table 8: Accuracy and time comparison on Single Document QA task of LongBench v1 (Bai et al., 2024).

|  | QA task acc (%) | Average Time |
|---|---|---|
| Llama-3.1-8B (baseline) | 25.4 | 1.0× |
| A3 (group size = 1) | **27.1** | 1.3× |
| A3 (group size = 2) | 22.5 | **0.7**× |

(group size 2) achieves  30% faster decoding, demonstrating that A3's groupwise inference can provide real latency benefits in longer contexts. This result shows that larger groups introduce more parallelism but can slightly reduce accuracy, which is consistent with our analyses in shorter contexts.

Our 8k experiments indicate that A3 can scale to significantly longer sequences without degradation and provides decoding-time advantages via groupwise generation. These results support the potential of A3 for future long-context extensions and we will explore longer context in future works.

