# OpenReview forum: "Autoregressive Models Rival Diffusion Models at ANY-ORDER Generation"
_ICLR.cc/2026/Conference — ICLR 2026 Poster_

### Official Review · Reviewer_ky8C · 2025-11-01

**Soundness:** 3
**Presentation:** 3
**Contribution:** 3
**Rating:** 6
**Confidence:** 3

**Summary:**

The paper proposes A3, a sequence modeling framework that generalizes AR factorization to predict arbitrary groups of tokens in any order, aiming to combine AR’s likelihood-faithfulness with the flexibility/parallelism of diffusion and semi-AR methods. The core is a two-stream attention design (content vs. query streams) plus a progressive curriculum (AR init → group expansion → order permutation). Experiments on QA/commonsense/story-infilling show A3 beats several diffusion LMs and scales with model size, though it still trails an AR baseline at equal parameter counts; the authors attribute this to smaller pretraining budgets.

**Strengths:**

- Neat architectural idea. The two-stream attention extends XLNet-style permutation to groups: content attends to ≤ current group; query attends only to prior groups, yielding a clean predictive head for the next group. The masks and equations are well specified.
- Thoughtful training curriculum. The three-stage progression (singleton AR → contiguous groups → permuted grouping) offers a practical path from a pretrained AR model to any-order behavior.
- Flexible inference recipes. The paper describes groupwise AR sampling and dynamic resampling, articulating a speed–quality trade-off and illustrating it empirically.
- Empirical signal vs. diffusion LMs. On TriviaQA/PIQA/etc., A3 (1B–8B) outperforms diffusion baselines and shows sensible scaling; the table also situates an AR baseline.

**Weaknesses:**

- Missing baselines from the semi-AR family. The related-work discussion notes multi-token prediction and insertion models, but the experiments do not include strong semi-autoregressive baselines (e.g.,  speculative-with-MTP) that directly target speed-ups while preserving AR quality. This makes it hard to attribute A3’s benefits to grouping vs. other established accelerations.
- Latency reporting is proxy-based. The speed-quality trade-off uses per-sequence decoding time and Llama-measured perplexity for unconditional generation; end-to-end wall-clock latency under realistic batching and KV-cache reuse is not reported for the task benchmarks. More concrete throughput/latency numbers (tokens/s, ms/sample) would help.
- Ablation depth. The method depends on group size schedules, permutation schemes, and the two-stream mask design. The paper lacks sensitivity analyses for (i) curriculum choices (stage lengths, s progression), (ii) group selection criteria during dynamic resampling (beyond brief entropy vs. confidence), and (iii) robustness across context lengths.

**Questions:**

- Compute-normalized comparison. Can you provide training FLOPs and data tokens for A3 vs. AR and diffusion baselines, and where possible train the AR baseline on the same 2B tokens to isolate the architectural effect?
- Semi-AR baselines. How does A3 compare against modern multi-token-prediction methods under matched setups? A small-scale controlled study would be informative.
- Curriculum sensitivity. What happens if you skip Stage 2 or change the group-size schedule? Is the two-stream architecture alone sufficient to get most of the gains?
- Inference consistency. For dynamic resampling, does there exist a well-defined induced factorization (e.g., an ordering over committed sets) that preserves AR semantics, or is this best viewed as a heuristic? Any failure cases?

---

> ### Author Response · Authors · 2025-11-24
> **Response to Reviewer ky8C (1/2)**
>
> We thank Reviewer ky8C for appreciating the neat idea and the thoughtful recipe of our new target. Below, we will address your concerns in the following points.
>
> ---
>
> **W1&Q2**. Missing baselines from the semi-AR family. How does A3 compare against modern multi-token-prediction methods under matched setups?
>
> **A1**. Thanks for your suggestions. We additionally conduct generation experiments as in Figure 3. We use speculative decoding with Llama-3.2-1B as the draft model and Llama-3.1-8B as the target model, matching A3’s group size (max 4). Perplexity is measured by Llama-3.1-70B. Results are shown below:
>
> |  | log(Perplexity) | Time |
> | --- | --- | --- |
> | Speculative decoding | 1.9 | 1.2 $\times$ |
> | A3 | 2.1 | 1 $\times$ |
>
> Speculative decoding achieves lower perplexity but at higher wall-clock cost, while A3 provides competitive quality with faster decoding in this setting.
>
> **Relationship between A3 and semi-AR methods.** Importantly, semi-AR methods such as speculative decoding and multi-token prediction are decoding-level accelerators: they maintain the same left-to-right AR factorization and improve efficiency through draft-model proposals or multi-step predictions. In contrast, A3 changes the model factorization itself by enabling groupwise and permutation-based prediction. This makes A3 **orthogonal** to speculative/MTP: these accelerations can also be applied on top of A3’s factorization in principle.
>
> ---
>
> **W2**. More concrete throughput/latency numbers under realistic batching and KV-cache reuse would help.
>
> **A2**. We agree that realistic latency including batching and KV-cache reuse is critical for evaluating practical speedups. Architecturally, A3 is fully compatible with standard KV-cache mechanisms: the two-stream attention and groupwise masks do not interfere with caching, and integrating KV reuse is straightforward. Our current implementation evaluates single-sample decoding without KV reuse, and we will include full batched KV-cached decoding in the camera-ready.
>
> To provide concrete latency numbers, we report wall-clock per-sample decoding time for the ROC story-filling task on a single A100 GPU. These numbers provide a consistent baseline for comparing A3 against AR and diffusion methods:
>
> |  | ROC filling | Latency number (s/sample) |
> | --- | --- | --- |
> | Llama-3.1-8B | 11.7/2.3/10.5 | 0.21 |
> | DiffuLlama | 23.3/5.5/21.2 | 0.17 |
> | A3-8B | 19.2/4.6/18.6 | 0.15 |
>
> A3 shows a **~33% latency reduction** compared to the AR baseline and a speed-performance trade-off compared to DiffuLlama in this setting. We will extend the latency study with KV-cached batching and include those results in the camera-ready version.
>
> ---
>
> **W3&Q3**. Ablation depth. The paper lacks sensitivity analyses for (i) curriculum choices, (ii) group selection criteria during dynamic resampling, and (iii) robustness across context lengths.
>
> **A3**. Thanks for your suggestions. We add additional sensitivity analyses and show the results below.
>
> **Curriculum choices.** We have explored different training curriculum, where we skip stage 1 and stage 2. We compare this training curriculum with the original one both training with 0.5B tokens. The results are shown in the following table.
>
> |  | TriQA | HSwag  | Wino.  | SIQA  | PIQA | ROCStories |
> | --- | --- | --- | --- | --- | --- | --- |
> | Original | 15.6 | 49.3 | 56.7 | 39.6 | 69.4 | 13.2/2.3/12.6 |
> | Skip Stage 1 & 2 | 11.3 | 44.2 | 54.1 | 37.3 | 64.2 | 13.1/2.2/12.4 |
>
> The model has non-trivial performance without the adaptation to new architecture (Stage 1 & 2), which indicates that two-stream architecture alone is sufficient to get most of the gains. However, it underperforms that with adaptation (the original one). The result shows the necessity of the adaptation in Stage 1 & 2.
>
> **Group selection criteria.** We study how the confidence threshold affects the performance in dynamic resampling. In the default setting, a group size of positions with highest confidence will be selected in each step. With threshold, the selected ones with lower confidence than threshold will be discarded (but still at least one will be selected). The results of generation perplexity are shown in the following table.
>
> |  | log(perplexity) | Time |
> | --- | --- | --- |
> | Default (with no threshold) | 2.32 | 1 $\times$ |
> | Threshold = 0.05 | 2.25 | 1.2 $\times$ |
> | Threshold = 0.1 | 2.25 | 1.7 $\times$ |
> | Threshold = 0.2 | 2.18 | 2.2 $\times$ |
>
> Larger threshold ensures generated tokens with higher quality, but at a cost of more decoding time.
>
> **Context length.** We conduct experiments with testing model loss on different context length. The results are shown in the following table.
>
> |  | model loss |
> | --- | --- |
> | Length=512 | 3.51 |
> | Length=1024 | 3.34 |
> | Length=2048 | 3.23 |
>
> The model loss keeps stable within the training length, indicating the robustness of A3 across different context lengths.
>
> We will add these discussions in the revised manuscript.

---

> ### Author Response · Authors · 2025-11-24
> **Response to Reviewer ky8C (2/2)**
>
> **Q1**. Compute-normalized comparison. Can you provide training FLOPs and data tokens for A3 vs. AR and diffusion baselines, and where possible train the AR baseline on the same 2B tokens to isolate the architectural effect?
>
> **A4**. **Training Budget.** Because most baselines do not report training FLOPs, we follow the standard estimate $6ND$ from Hoffmann et al. [1], where $N$ is the number of parameters and $D$ is the number of training tokens. The approximate budgets for 7–8B models are:
>
> |  | Training FLOPs | Data tokens |
> | --- | --- | --- |
> | Llama-3.1-8B | ~7*10^23 | 15T |
> | Dream | ~4*10^22 | 0.6T |
> | DiffuLlama | ~4*10^21 | 0.06T |
> | A3-8B | ~2*10^20 | 0.002T |
>
> Despite using much fewer tokens and FLOPs, A3 matches or surpasses diffusion-style baselines. This underscores the high data efficiency of A3 as claimed in Section 3.2.
>
> **Isolating architecture effect.** We agree that ideally one would compare A3 to an AR model trained on the same 2B tokens. However, this comparison is **not symmetric**, because the 2B-token budget plays fundamentally different roles for the two models:
>
> **For AR**, the extra 2B tokens simply continue optimizing the *same* left-to-right factorization that was trained on hundreds of billions of tokens.
>
> **For A3**, the same 2B tokens must instead teach the model an entirely *new* factorization—groupwise prediction, permutation masks, and any-order conditioning. This is a substantially harder learning objective than more AR pretraining.
>
> Thus, comparing A3 directly against a fully pretrained AR model does not measure architectural differences. A fair test is to track how A3 improves under increasing post-training data.
>
> |  | TriviaQA | log(Perplexity) |
> | --- | --- | --- |
> | A3 (1.5B tokens) | 16.2 | 2.9 |
> | A3 (2B tokens) | 19.4 | 2.5 |
> | A3 (2.5B tokens) | 22.5 | 2.3 |
> | AR | 52.1 | 0.8 |
>
> Performance increases steadily with more data. This confirms that A3 benefits strongly from data scale and that the gap to fully-trained AR models is due to training budget, not a limitation of the A3 architecture. We will add this clarification in the revised paper.
>
> [1] J.Hoffmann et al. Training Compute-Optimal Large Language Models. arXiv:2203.15556
>
> ---
>
> **Q4**. Inference consistency. For dynamic resampling, does there exist a well-defined induced factorization (e.g., an ordering over committed sets) that preserves AR semantics, or is this best viewed as a heuristic? Any failure cases?
>
> **A5**. Dynamic resampling in A3 should be viewed as a practical inference heuristic rather than a provably distribution-correct sampling algorithm. Each update step is consistent with the A3 factorization—i.e., every committed subset corresponds to a valid conditional term
>
> $P(x_{G_t} \mid x_{\cup_{j < t} G_j})$
>
> but the sequence of updates is chosen adaptively based on confidence or entropy, and therefore does not correspond to a single pre-defined global ordering over all groups. In this sense, dynamic resampling provides **locally AR-consistent updates.**
>
> Empirically, we find that dynamic resampling behaves stably across tasks. It typically does not exhibit oscillation or divergence. The main failure mode appears when very large groups are sampled early in generation; this can temporarily reduce confidence across many positions and, in rare cases, cause further degradation. Using small group sizes or higher confidence thresholds prevents such issues.
>
> Overall, dynamic resampling is designed as a **practical, scalable inference strategy** that maintains local AR semantics while enabling flexibility and speed. We will add this discussion in the paper.
>
> ---
>
> Thanks again for your detailed comments, which are very helpful, and hope our response could address your concerns. Please let us know if you have additional questions.

---

### Official Review · Reviewer_yNMv · 2025-11-02

**Soundness:** 3
**Presentation:** 2
**Contribution:** 2
**Rating:** 4
**Confidence:** 2

**Summary:**

The paper proposes A3 (Any-order Any-subset Autoregressive), a sequence modeling framework that keeps the probabilistic rigor and training stability of standard autoregressive (AR) models but relaxes the decoding order to arbitrary groups of tokens. Concretely, A3 factorizes a sequence into groups and trains the model to predict any group conditioned on previously generated groups, using a two-stream attention layout to separate “what has been generated” from “what is now being predicted.” At inference, A3 can decode group-by-group (for speed) or do dynamic re-sampling of uncertain positions (for quality), so it can handle infilling and other non-left-to-right scenarios in a single model. Empirically, on common LM benchmarks (QA, commonsense, story cloze), A3 outperforms discrete-diffusion / masked-iterative baselines at similar or larger compute, but still trails a comparable AR model that was trained on far more tokens.

**Strengths:**

1. Clear target: AR-level stability + arbitrary-order generation. The paper correctly identifies a real and currently active gap: classic AR models are stable and likelihood-faithful but order-rigid, while discrete diffusion / iterative masked LM are order-flexible but multi-step, hyperparameter-sensitive, and sometimes harder to train. Proposing a single framework that “looks like AR to the optimizer” but “behaves like an infiller” is a sensible and timely goal.
2. Groupwise factorization is a clean, composable idea. Writing ($p(x)=\prod_k p(x_{G_k}\mid x_{G_{<k}})$) and randomizing the grouping/order during training is conceptually simple, keeps likelihood well-defined, and reuses the well-understood AR training pipeline. Compared with diffusion-style schedules, this reduces the number of design knobs while still giving order flexibility.
3. Two-stream attention is a reasonable architectural choice. Reusing an XLNet-style separation between “content” and “query” to implement “see only past groups but write to current group” is a sensible reuse of an established mechanism, so the method is not just algorithmic but also implementable on existing Transformer stacks.

**Weaknesses:**

1. Parallel generation is only *heuristically* correct, not *distributionally* correct. The paper’s decoding story (“decode some groups, resample the uncertain ones”) is an engineering compromise, but it does not give the kind of provable, joint-distribution-correct parallel sampling that very recent any-subset AR work is starting to provide (e.g. ASSD in Guo & Ermon 2025) — those works explicitly address the mismatch between parallel predictions and the target joint, while A3 largely sidesteps it. This makes the “any-subset” claim weaker on the sampling-theoretic dimension.
2. Comparisons are not load-bearing for the main claim. The paper leans on the fact that it beats discrete/diffusion-style baselines, but those baselines are known to be data/step/hyperparameter hungry; meanwhile A3 is compared to a much better funded AR line only anecdotally (“we used fewer tokens, so we lag”). Without an equal-data, equal-backbone, equal-context comparison against a strong AR infiller (e.g. a repurposed XLNet / permutation LM, or an AR+speculative multi-token decoder), it is hard to measure how much of the gain comes from the grouping idea itself vs. from simply staying in the AR training regime.
3. Method novelty is incremental over permutation-style / two-stream LMs. At a high level, A3 = permutation/pseudo-permutation LM (XLNet-like) + explicit grouping curriculum + a decoding heuristic. The paper frames it as a “generalization of AR to any order, any subset,” but several of the enabling ingredients — two-stream separation, masked/pseudo-permutation training, curriculum over masks — have been explored in earlier LM or masked-hard-attention work. The paper’s specific combination is tidy, but the conceptual leap is smaller than the title suggests.
4. Group choice is exogenous and may be the hard part. The model is trained on random / curriculum-defined partitions, but the real deployments that need “any subset” (layout-controlled editing, multi-span infilling, multi-document patching) tend to have structured subsets (spans aligned to discourse / layout). A3 doesn’t learn the grouping policy, and the paper doesn’t show that the learned model is robust to strongly biased or adversarial group patterns. That is exactly where arbitrary-order models tend to break.

**Questions:**

1. On distributional correctness: your dynamic re-sampling is essentially a confidence- or entropy-driven iterative refinement. How do you ensure that, after several such rounds, the final joint over all tokens is still close to the model’s intended factorization — and how does this compare empirically to ASSD-style “correct-by-construction” decoding for any-subset ARMs? (A small experiment against the scheme in Guo & Ermon 2025 would make the generative claim much harder to dispute.)
2. On asymptotic scaling vs. plain AR: you attribute the quality gap to “only 2B tokens,” but you also inject extra supervision signals (more factorization patterns, more masked positions). Do you have evidence that, at equal data and wall-clock, A3 does not hurt the base AR’s next-token perplexity, especially on long contexts where order permutations are most disruptive? A curve like “data ↑ → (AR PPL, A3 PPL) → gap ↓” would make the core claim much more convincing.

---

> ### Author Response · Authors · 2025-11-24
> **Response to Reviewer yNMv (1/3)**
>
> We thank Reviewer yNMv for appreciating the our formulation and realization of the new target. Below, we will address your concerns in the following points.
>
> ---
>
> **W1&Q1**. On distributional correctness: your dynamic re-sampling is essentially a confidence- or entropy-driven iterative refinement. How do you ensure that, after several such rounds, the final joint over all tokens is still close to the model’s intended factorization — and how does this compare empirically to ASSD-style “correct-by-construction” decoding for any-subset ARMs? (A small experiment against the scheme in Guo & Ermon 2025 would make the generative claim much harder to dispute.)
>
> **A1**. We appreciate your finding on the parallel generation. We will explain it in two aspects.
>
> > How do you ensure that, after several such rounds, the final joint over all tokens is still close to the model’s intended factorization?
>
> In fact, each update step is consistent with the A3 factorization—i.e., every committed subset corresponds to a valid conditional term $P(x_{G_t} \mid x_{\cup_{j < t} G_j})$, but the sequence of updates $x_{G_t}$ is chosen adaptively based on confidence or entropy. Although this does not correspond to a single pre-defined global ordering over all groups, however, in this sense, dynamic resampling provides **locally AR-consistent updates.** This makes the final joint over all tokens still close to the model’s semi-autoregressive factorization, but the group partitions may not be sampled from uniform distribution as in training — This is a good thing in fact. We can sample a group partition with higher quality using dynamic resampling, without breaking the model’s factorization.
>
> > How does this compare empirically to ASSD-style “correct-by-construction” decoding for any-subset ARMs?
> >
>
> The motivation for ASSD using correct-by-construction decoding is that, they model the sequence one token by one token using absorning state DTMC and assume
>
>  $\sum_{i \in [m, N)} \log p(x_{\sigma(i)} \mid \mathbf{x}\_{\sigma(<m)}) \neq \log p(\mathbf{x}\_{\sigma(\geq m)} \mid \mathbf{x}\_{\sigma(<m)})$
>
> Therefore, when they sample a new group $\sigma(m),\dots,\sigma(N-1)$, they need to use rejection sampling to get the right distribution for the new group. However, A3 directly models the sequence group by group. Therefore, the sampling results from $P(x_{G_t} \mid x_{\cup_{j < t} G_j})$ in each step faithfully represent the true distribution.
>
> We compare A3’s dynamical resampling with confidence and ASSD sampling method in unconditional generation as the same setting in Figure 3 using a group size of 4. We show the results in the following table:
>
> |  | Perplexity | Time |
> | --- | --- | --- |
> | A3 | 2.2 | 1 $\times$ |
> | ASSD | 2.3 | 2.4 $\times$ |
>
> With comparative results on generation quality, ASSD costs 2.4$\times$ time due to additional computation for resampling. This proves the high quality and high efficiency of A3’s dynamic resampling method.
>
> ---

---

> ### Author Response · Authors · 2025-11-24
> **Response to Reviewer yNMv (2/3)**
>
> **W2&Q2**. Without an equal-data, equal-backbone, equal-context comparison against a strong AR infiller (e.g. a repurposed XLNet / permutation LM, or an AR+speculative multi-token decoder), it is hard to measure how much of the gain comes from the grouping idea itself vs. from simply staying in the AR training regime. Do you have evidence that, at equal data and wall-clock, A3 does not hurt the base AR’s next-token perplexity, especially on long contexts where order permutations are most disruptive? A curve like “data ↑ → (AR PPL, A3 PPL) → gap ↓” would make the core claim much more convincing.
>
> **A2.** **AR variants baseline.** We appreciate the reviewer’s suggestion. We additionally conduct generation experiments as in Figure 3. We use speculative decoding with Llama-3.2-1B as the draft model and Llama-3.1-8B as the target model, matching A3’s group size (max 4). Perplexity is measured by Llama-3.1-70B. Results are shown below:
>
> |  | log(Perplexity) | Time |
> | --- | --- | --- |
> | Speculative decoding | 1.9 | 1.2 $\times$ |
> | A3 | 2.1 | 1 $\times$ |
>
> Speculative decoding achieves lower perplexity but at higher wall-clock cost, while A3 provides competitive quality with faster decoding in this setting.
>
> Importantly, semi-AR methods such as speculative decoding and multi-token prediction are decoding-level accelerators: they maintain the same left-to-right AR factorization and improve efficiency through draft-model proposals or multi-step predictions. In contrast, A3 changes the model factorization itself by enabling groupwise and permutation-based prediction. This makes A3 **orthogonal** to speculative/MTP: these accelerations can also be applied on top of A3’s factorization in principle.
>
> **Performance with more data.** We track how A3 improves under increasing post-training data.
>
> |  | TriviaQA | log(Perplexity) |
> | --- | --- | --- |
> | A3 (1.5B tokens) | 16.2 | 2.9 |
> | A3 (2B tokens) | 19.4 | 2.5 |
> | A3 (2.5B tokens) | 22.5 | 2.3 |
> | AR | 52.1 | 0.8 |
>
> Performance increases steadily with more data. This confirms that A3 benefits strongly from data scale and that the gap to fully-trained AR models is due to training budget, not a limitation of the A3 architecture.
>
> **Long-context ability.** To evaluate whether A3 hurt the base AR’s next-token prediction ability under long contexts, we finetuned both Llama-3.1-8B and A3-8B on 8k-length sequences from PG19 using the same training budget (100 steps, batch size 64). We then evaluated them on the **single-document QA task** from LongBench v1 [1], which requires reasoning over long passages. We used dynamic sampling for A3 with group size 1 and group size 2.
>
> |  | QA task acc (\%) | Average Time |
> | --- | --- | --- |
> | Llama 3.1 8B (baseline) | 25.4 | 1.0$\times$  |
> | A3 (group size = 1) | 27.1 | 1.3$\times$  |
> | A3 (group size = 2) | 22.5 | 0.7$\times$  |
>
> A3 (group size 1) improves over the AR baseline with 1.7\%, suggesting that the A3 factorization does not degrade long-context modeling and may offer small gains without parallel decoding. A3 (group size 2) achieves ~30% faster decoding, demonstrating that A3’s groupwise inference can provide real latency benefits in longer contexts. This result shows that larger groups introduce more parallelism but can slightly reduce accuracy, which is consistent with our analyses in shorter contexts. These experiments indicate that A3 can scale to long context without degradation.
>
> [1] Bai et al. LongBench: A Bilingual, Multitask Benchmark for Long Context Understanding. ACL 2024

---

> ### Author Response · Authors · 2025-11-24
> **Response to Reviewer yNMv (3/3)**
>
> **W3**. Method novelty is incremental over permutation-style / two-stream LMs. At a high level, A3 = permutation/pseudo-permutation LM (XLNet-like) + explicit grouping curriculum + a decoding heuristic. The paper frames it as a “generalization of AR to any order, any subset,” but several of the enabling ingredients — two-stream separation, masked/pseudo-permutation training, curriculum over masks — have been explored in earlier LM or masked-hard-attention work.
>
> **A3**. Thanks for pointing this out. We totally agree that A3 leans on well-established building blocks (two-stream attention, permutation training, curriculum learning). But we want to clarify that the way we’ve combined these tools *with a sharp focus on generative tasks*, and the capabilities that result, are far from just small, incremental tweaks.
>
> Our main goal here isn’t to cook up brand-new components. Instead, it’s to bring AR modeling into any-order generation to expand its **generative** ability—a huge gap that previous frameworks have left wide open. This unification leads to three outcomes that go way beyond “just combining tools”:
>
> 1. **Built for Generation, Not Just Understanding**
> Prior work on two-stream attention or permutation (like XLNet) was all about boosting contextual understanding like hink QA and text classification. A3 repurposes these tools to do something AR models never could: **generate text in any order, or even just specific subsets of it**. And it does this while hanging onto AR’s biggest strengths—solid probabilistic rigor and stable training, which diffusion models have to sacrifice.
>
> 2. **Scales to Big Models for Generative Work**
> A3 is the first to take permutation-based AR and scale it up to **7B+ parameter models** for generative tasks. The earlier ideas behind permutation LMs (like the core concept in XLNet) never got past single-token prediction. It’s A3's group-based curriculum and decoding strategies that make large-scale, parallel, and flexible generation actually work.
>
> 3. **Turns AR Into a Flexible Baseline**
> By proving AR can break free from left-to-right ordering, A3 delivers a key message: AR models—with their rock-solid training stability and high sample quality—don’t have to be tossed aside for diffusion when flexibility is needed. It redefines what AR can do in any-order generative modeling.
>
> To sum it up: A3’s novelty isn’t in new parts—it’s in how we’ve repurposed and unified old ones to **unlock AR’s hidden generative flexibility**. No prior framework has addressed that gap. The real value is that we open up a whole new direction for any-order generation.
>
> ---
>
> **W4**. Group choice is exogenous and may be the hard part. The model is trained on random / curriculum-defined partitions, but the real deployments that need “any subset” (layout-controlled editing, multi-span infilling, multi-document patching) tend to have structured subsets (spans aligned to discourse / layout). A3 doesn’t learn the grouping policy, and the paper doesn’t show that the learned model is robust to strongly biased or adversarial group patterns. That is exactly where arbitrary-order models tend to break.
>
> **A4**. Indeed, A3 does not explicitly learn a grouping policy. Our goal in this work is to design an any-order factorization with autoregressive modeling, not to optimize group selection itself. In fact, A3 is designed to operate correctly under **any externally supplied partition:** Although A3 is trained on random and curriculum-based partitions, the underlying mechanism can generalize well to structured subsets. The two-stream attention and groupwise causal masks ensure that the model always conditions on committed tokens and predicts uncommitted ones regardless of how the subset is shaped.
>
> Additionally, dynamic sampling at inference provides a practical way to iteratively refine uncertain regions while respecting the A3 factorization. That means the model can inference which next group to generate is better in each generation step. This allows the model to adjust to irregular or highly imbalanced subsets without breaking consistency.
>
> We agree that learning grouping policies is an interesting extension—e.g., jointly optimizing content selection and sequence construction. But we view it as a complementary research direction rather than a missing piece of A3’s core contribution. We will explore this in the future work.
>
> ---
>
> Thanks again for your detailed comments, which are very helpful, and hope our response could address your concerns. Please let us know if you have additional questions.

---

### Official Review · Reviewer_TFMj · 2025-11-02

**Soundness:** 2
**Presentation:** 3
**Contribution:** 2
**Rating:** 2
**Confidence:** 4

**Summary:**

The paper addresses a challenge in sequence generation: reconciling the training stability/ generation quality of autoregressive (AR) models with the flexibility/ parallelism of diffusion-based models. This problem is theoretically meaningful, as existing paradigms struggle to balance efficiency, quality, and adaptability—key requirements for real-world applications like long-context reasoning, text infilling, and fast content generation.

A3 resolves this by generalizing AR factorization to support "groupwise token prediction in arbitrary orders," with three core designs:
(1) two-stream attention architecture, (2) progressive training strategy, and (3) flexible inference, which supports groupwise AR sampling (fast, fixed group sizes) and dynamic resampling (high-quality, confidence/entropy-based subset selection).

Empirically, A3 (trained on 2B tokens) outperforms state-of-the-art diffusion models across QA (TriviaQA), commonsense reasoning (PIQA, HellaSwag), and infilling (ROCStories) tasks (e.g., A3-8B achieves 78.1% PIQA accuracy vs. 63.3% for DiffuLlama-7B). It also surpasses AR models in AR-disadvantaged tasks (e.g., ROCStories ROUGE-L: 18.6 vs. 10.5 for Llama-3.1-8B).

**Strengths:**

1. The paper focus a critical problem, which addresses a gap in sequence generation—reconciling AR’s stability/quality with diffusion’s flexibility/parallelism—aligned with real-world needs (long-context generation, infilling).

2. Explicitly outperforms AR models in infilling (ROCStories) by utilizing bidirectional context, validating its flexibility.

**Weaknesses:**

1. Fails to cite or discuss existing NAR research on Block-wise generation (e.g., Block-AR models that split sequences into fixed/masked blocks for parallel prediction) and progressive training (e.g., curriculum-based NAR training that increments block size or relaxes order constraints). This gap obscures A3’s incremental innovation—readers cannot distinguish whether A3’s groupwise/progressive designs are novel or iterative improvements on prior NAR work.

2. No controlled experiment comparing A3 with an AR model trained on the same 2B tokens (e.g., fine-tuned Llama-3.1-8B). The gap with top AR models (e.g., 19.4% vs. 52.1% on TriviaQA) may stem from data scarcity, not A3’s design—undermining assessment of its architectural advantage.

2. Only evaluates sequences of length 2048, with no tests on long contexts (16k/128k tokens). A3’s claimed advantage in solving AR’s long-context inefficiency remains unproven.

3. Figure 3 claims 3–4x faster decoding than AR but lacks direct time data and performances for baseline AR models (same hardware/sequence length) and explicit absolute time values, making efficiency gains hard to quantify.

**Questions:**

See weaknesses.

---

> ### Author Response · Authors · 2025-11-24
> **Response to Reviewer TFMj (1/2)**
>
> We thank Reviewer TFMj for appreciating the importance of our research and the superior performance. Below, we will address your concerns in the following points.
>
> ---
>
> **Q1**. Fails to cite or discuss existing NAR research on Block-wise generation and progressive training. This gap obscures A3’s incremental innovation—readers cannot distinguish whether A3’s groupwise/progressive designs are novel or iterative improvements on prior NAR work.
>
> **A1**. Thanks so much for your thoughtful feedback—we really appreciate getting to clarify that the NAR work you mentioned (Block-wise generation and curriculum-based progressive training) is already cited and detailed in **Appendix B** of the original manuscript. Here’s the specific breakdown:
>
> - Block-wise generation gets covered in the "Multi-token Prediction" subsection—we cite Stern et al., 2018’s blockwise decoding there—and also in the "Non-autoregressive Generation" section.
> - For curriculum-based NAR training, we discuss it in the "Non-autoregressive Generation" subsection (citing Guo et al., 2020) and the "Continue Pre-training" section too.
>
> On the point of improvements on prior NAR work, we want to stress that A3’s core goal isn’t to make incremental improvements to existing Block-AR or curriculum-based NAR methods. Instead, it’s meant to **bring the autoregressive (AR) framework back into the "any-order generation" space**—an area that’s mostly been taken over by diffusion models lately. It’s not competing directly with traditional AR or small tweaks to NAR. Here is the key difference that makes A3’s novelty clear:
>
> 1. Focusing Paradigm: The paradigms we’re focusing on are different. Existing Block-AR or NAR methods are all about optimizing fixed-structure parallelism to speed things up. However, A3’s achieves breakthrough by also showing AR can be expanded to handle any-order generation, which lets us bring AR’s strengths—like rigorous probability modeling and stable training—to a space where diffusion used to be the only option.
> 2. Accelerating ability: Existing block-wise/multi-token prediction approaches typically: 1. depend on multiple linear heads, which limits maximum parallelism, 2. or speculative decoding, where efficiency is bounded by the draft model’s quality. Therefore, they cannot reach the **same theoretical decoding parallelism** as diffusion-style iterative refinement. A3, by contrast, can **reuse diffusion-like scheduling** using a single AR model, enabling it to achieve a higher theoretical upper bound on parallel decoding efficiency while remaining within an AR framework.
>
> We will add this discussion in the revised version.
>
> ---
>
> **Q2**. No controlled experiment comparing A3 with an AR model trained on the same 2B tokens (e.g., fine-tuned Llama-3.1-8B). The gap with top AR models (e.g., 19.4% vs. 52.1% on TriviaQA) may stem from data scarcity, not A3’s design—undermining assessment of its architectural advantage.
>
> **A2**. We agree that the performance gap is primarily due to data scale, but it is important to note that A3 and AR use the 2B-token budget for fundamentally different purposes, so comparing them under the same 2B tokens is *not* symmetric.
>
> **For AR**, the extra 2B tokens simply continue optimizing the *same* left-to-right factorization that was trained on hundreds of billions of tokens.
>
> **For A3**, the same 2B tokens must instead teach the model an entirely *new* factorization—groupwise prediction, permutation masks, and any-order conditioning. This is a substantially harder learning objective than more AR pretraining.
>
> Thus, comparing A3 to a fully-pretrained AR model does not isolate architectural effects
>
> To illustrate that the remaining gap is scale-related rather than architectural, we show how A3 improves as we increase the post-training data:
>
> |  | TriviaQA | log(Perplexity) |
> | --- | --- | --- |
> | A3 (1.5B tokens) | 16.2 | 2.9 |
> | A3 (2B tokens) | 19.4 | 2.6 |
> | A3 (2.5B tokens) | 22.5 | 2.4 |
> | AR | 52.1 | 0.8 |
>
> Performance increases steadily with more data. This confirms that A3 benefits strongly from data scale and that the gap to fully-trained AR models is due to training budget, not a limitation of the A3 architecture. We will clarify this distinction in the revised paper.

---

> ### Author Response · Authors · 2025-11-24
> **Response to Reviewer TFMj (2/2)**
>
> **Q3**. Only evaluates sequences of length 2048, with no tests on long contexts (16k/128k tokens).
>
> **A3**. We appreciate the reviewer’s point. Indeed, A3 ’s groupwise factorization and parallel decoding could help mitigate some of the practical inefficiencies of left-to-right decoding at longer sequence lengths, and we provide preliminary evidence toward this.
>
> ### **Experimental study on 8k-length sequences**
>
> To evaluate whether A3 remains stable under significantly longer contexts than 2k tokens, we finetuned both Llama-3.1-8B and A3-8B on 8k-length sequences from PG19 using the same training budget (100 steps, batch size 64). We then evaluated them on the **single-document QA task** from LongBench v1 [2], which requires reasoning over long passages. We used dynamic sampling for A3 with group size 1 and group size 2.
>
> |  | QA task acc (\%) | Average Time |
> | --- | --- | --- |
> | Llama 3.1 8B (baseline) | 25.4 | 1.0$\times$  |
> | A3 (group size = 1) | 27.1 | 1.3$\times$  |
> | A3 (group size = 2) | 22.5 | 0.7$\times$  |
>
> A3 (group size 1) improves over the AR baseline with 1.7\%, suggesting that the A3 factorization does not degrade long-context modeling and may offer small gains without parallel decoding. A3 (group size 2) achieves ~30% faster decoding, demonstrating that A3’s groupwise inference can provide real latency benefits in longer contexts. This result shows that larger groups introduce more parallelism but can slightly reduce accuracy, which is consistent with our analyses in shorter contexts.
>
> Our 8k experiments indicate that A3 can scale to significantly longer sequences without degradation and provides decoding-time advantages via groupwise generation. These results support the potential of A3 for future long-context extensions and we will explore longer context in future works.
>
> ---
>
> **Q4**. Figure 3 claims 3–4x faster decoding than AR but lacks direct time data and performances for baseline AR models (same hardware/sequence length) and explicit absolute time values.
>
> **A4**. Thanks for your suggestions. We now explicitly measure decoding time for Llama-3.1-8B (AR baseline) under the same hardware (A100), same sequence length (2048 tokens), and same batch size as the A3 experiments in Figure 3. We evaluate all models’ log-perplexity with Llama-3.1-70B.
>
> |  | log(Perplexity) | Time |
> | --- | --- | --- |
> | Llama-3.1-8B (baseline) | 1.9 | 67s |
> | A3 (group size = 1) | 1.7 | 142s |
> | A3 (group size = 2) | 1.8 | 71s |
> | A3 (group size = 4) | 2.1 | 37s |
>
> With moderate groups (size 4), A3 achieves faster decoding than the AR baseline (67s $\to$ 37s) at a small quality tradeoff. This proves A3’s decoding efficiency.
>
> ---
>
> Thanks again for your detailed comments, which are very helpful, and hope our response could address your concerns. Please let us know if you have additional questions.

---

### Official Review · Reviewer_6bME · 2025-11-02

**Soundness:** 3
**Presentation:** 3
**Contribution:** 3
**Rating:** 6
**Confidence:** 3

**Summary:**

This paper introduces A³ (Any-order Any-subset Autoregressive modeling), a generalization of standard autoregressive (AR) language modeling. Instead of predicting tokens strictly left-to-right, A³ allows generation in arbitrary orders and subsets while maintaining a valid probabilistic formulation.

The method builds on a two-stream attention mechanism (content/query streams) similar to XLNet and uses a curriculum training schedule that progressively transitions from left-to-right to multi-token and random-order prediction. At inference, A³ supports both groupwise parallel decoding and dynamic resampling, trading off speed and quality.

Experiments across QA, reasoning, and infilling tasks show that A³ matches or surpasses diffusion-based models (e.g., DiffuLlama, Dream) using significantly less data, and maintains competitive performance to AR models while enabling faster, flexible generation.

Overall it is a practical step toward bridging AR and diffusion paradigms, retaining AR stability while introducing parallel generation flexibility.

**Strengths:**

1. Insightful formulation. The paper presents a very interesting perspective on any-order, any-subset autoregressive modeling, effectively bridging the strengths of AR and masked diffusion models in a unified probabilistic framework.

2. Clarity and organization. The paper is well written and easy to follow, with clear explanations, sound motivation, and well-structured methodology.

3. Experiments across multiple reasoning and generation benchmarks validate the effectiveness of the proposed approach, showing competitive or superior performance with improved flexibility.

**Weaknesses:**

1. What's new relative to prior AR generalizations? The author mentioned that the proposed method builds closely on existing ideas from XLNet (permutation-based AR) and masked diffusion modeling, with the main difference being a unified training/inference view. While conceptually elegant, the contribution may feel incremental rather than fundamentally new.

2. Evaluation scope and ablations are limited. Experiments mainly compare against diffusion-style baselines; there is less analysis against modern AR variants (e.g., speculative decoding, parallelized transformers) or ablations on key design choices like grouping strategy and curriculum schedule.

3. Practical speed–quality trade-offs unclear. Although the paper claims parallel generation benefits, detailed runtime comparisons and latency measurements are missing, making it hard to assess the real efficiency gains of “any-order” decoding in practice.

**Questions:**

1. How does A³ fundamentally differ from prior permutation-based autoregressive approaches such as XLNet or Permutation LM beyond the introduction of multi-token subsets? Could you clarify what new modeling capacity A³ enables that these earlier frameworks cannot?

2. The paper emphasizes that A³ enables parallel or groupwise decoding. Could you provide concrete runtime or latency benchmarks (e.g., decoding speedups vs. standard AR and diffusion models) to quantify the practical benefit of this flexibility?

3. The curriculum that transitions from left-to-right to any-order prediction seems important. How sensitive is model performance to the curriculum schedule (e.g., ratio of L2R vs. random-order batches)? Have you explored automatically learned or adaptive curricula?]

4. Since any-order generation can, in principle, apply to structured data (e.g., image or audio tokens), do you anticipate A³ extending naturally to multimodal diffusion–AR hybrids, or would significant architectural adjustments be required?

---

> ### Author Response · Authors · 2025-11-24
> **Response to Reviewer 6bME (1/2)**
>
> We thank Reviewer 6bME for appreciating our insightful formulation, clear organization and superior performance. Below, we will address your concerns in the following points.
>
> ---
>
> **W1&Q1.** What's new relative to prior AR generalizations? The author mentioned that the proposed method builds closely on existing ideas from XLNet (permutation-based AR) and masked diffusion modeling, with the main difference being a unified training/inference view. While conceptually elegant, the contribution may feel incremental rather than fundamentally new. How does A3 fundamentally differ from prior permutation-based autoregressive approaches such as XLNet or Permutation LM beyond the introduction of multi-token subsets? Could you clarify what new modeling capacity A3 enables that these earlier frameworks cannot?
>
> **A1.** Thank you for highlighting this important distinction. To clarify how A3 differs from prior work like XLNet: XLNet relies on a token-level permutation objective—randomly reordering individual tokens for sequential prediction—yet it remains tied to the single-token prediction paradigm, even with its departure from left-to-right ordering. In contrast, A3 introduces a group-level prediction framework: we partition sequences into arbitrary token groups and support flexible prediction of any subsets in any order. This design enables parallel decoding of multiple tokens simultaneously, directly addressing the efficiency bottleneck of token-wise generation—a limitation XLNet does not overcome.
>
> To summarize, A3 is not simply an extension of XLNet’s token-level permutation concept. Instead, it elevates the modeling granularity to token groups, enabling a key breakthrough: moving from “single-token arbitrary order prediction” to “multi-token arbitrary subset/order generation.” This advancement stems from architectural optimizations and training innovations, while preserving the probabilistic rigor and training stability of autoregressive models. Critically, this addresses core requirements of generative tasks—an area XLNet does not cover, as its focus lies on enhancing contextual understanding for comprehension-focused tasks.
>
> ---
>
> **W2&Q3.** Evaluation scope and ablations are limited. Experiments mainly compare against diffusion-style baselines; there is less analysis against modern AR variants (e.g., speculative decoding, parallelized transformers) or ablations on key design choices like grouping strategy and curriculum schedule. The curriculum that transitions from left-to-right to any-order prediction seems important. How sensitive is model performance to the curriculum schedule (e.g., ratio of L2R vs. random-order batches)? Have you explored automatically learned or adaptive curricula?]
>
> **A2.** **AR variants baseline**. We appreciate the reviewer’s suggestion. We additionally conduct generation experiments as in Figure 3. We use speculative decoding with Llama-3.2-1B as the draft model and Llama-3.1-8B as the target model, matching A3’s group size (max 4). Perplexity is measured by Llama-3.1-70B. Results are shown below:
>
> |  | log(Perplexity) | Time |
> | --- | --- | --- |
> | Speculative decoding | 1.9 | 1.2 $\times$ |
> | A3 | 2.1 | 1 $\times$ |
>
> Speculative decoding achieves lower perplexity but at higher wall-clock cost, while A3 provides competitive quality with faster decoding in this setting.
>
> Importantly, semi-AR methods such as speculative decoding and multi-token prediction are decoding-level accelerators: they maintain the same left-to-right AR factorization and improve efficiency through draft-model proposals or multi-step predictions. In contrast, A3 changes the model factorization itself by enabling groupwise and permutation-based prediction. This makes A3 **orthogonal** to speculative/MTP: these accelerations can also be applied on top of A3’s factorization in principle.
>
> *(A2 to be continued)*

---

> ### Author Response · Authors · 2025-11-24
> **Response to Reviewer 6bME (2/2)**
>
> *(A2 continued)
>
> **Curriculum schedule**. We agree that the curriculum is important. A3 introduces a **different causal mask and attention flow** from a standard AR transformer, and the model must progressively adapt from strict left-to-right prediction to multi-token and eventually arbitrary-order factorization. This mirrors curriculum strategies used in previous diffusion iterative refinement models [1].
>
> To assess sensitivity, we trained two variants on 0.5B tokens:
>
> 1. **Original curriculum** (L2R → groups → random permutations).
> 2. **Skipping Stage 1 and 2** (directly training on random permutations).
>
> The results are:
>
> |  | TriQA | HSwag  | Wino.  | SIQA  | PIQA | ROCStories |
> | --- | --- | --- | --- | --- | --- | --- |
> | Original | 15.6 | 49.3 | 56.7 | 39.6 | 69.4 | 13.2/2.3/12.6 |
> | Skip Stage 1 & 2 | 11.3 | 44.2 | 54.1 | 37.3 | 64.2 | 13.1/2.2/12.4 |
>
> Skipping the early stages consistently hurts performance by 4–6 points on several benchmarks, which proves the importance of such adaptation stage. Although we did not explore learned curricula in this work, we agree the idea is compelling. An adaptive schedule, e.g., based on training loss, may further improve robustness. We plan to investigate this direction in the future work.
>
> [1] Gong et al. Scaling Diffusion Language Models via Adaptation from Autoregressive Models. ICLR 2025
>
> ---
>
> **W3&Q2**. Practical speed–quality trade-offs unclear. Although the paper claims parallel generation benefits, detailed runtime comparisons and latency measurements are missing, making it hard to assess the real efficiency gains of “any-order” decoding in practice. The paper emphasizes that A3 enables parallel or groupwise decoding. Could you provide concrete runtime or latency benchmarks (e.g., decoding speedups vs. standard AR and diffusion models) to quantify the practical benefit of this flexibility?
>
> **A3**. Thanks for your suggestions. We now explicitly measure decoding time for Llama-3.1-8B (AR baseline) and DiffuLlama (Diffusion baseline) under the same setting in Figure 3. We evaluate all models’ log-perplexity with Llama-3.1-70B.
>
> |  | log(Perplexity) | Time |
> | --- | --- | --- |
> | Llama-3.1-8B (baseline) | 1.9 | 67s |
> | DiffuLlama (group size = 1) | 1.9 | 102s |
> | DiffuLlama (group size = 2) | 2.2 | 51s |
> | DiffuLlama (group size = 4) | 2.3 | 25s |
> | A3 (group size = 1) | 1.7 | 142s |
> | A3 (group size = 2) | 1.8 | 71s |
> | A3 (group size = 4) | 2.1 | 37s |
>
> Compared with AR baseline, with small groups (size 1 & 2), A3 achieves better performance at the trade of longer time due to more complex architecture. With moderate groups (size 4), A3 achieves faster decoding than the AR baseline (67s $\to$ 37s) at a small quality tradeoff. Comapred with diffusion baseline, A3 consistently performs better with the same group size or with the same time (e.g. A3 2.1 37s v.s. DiffuLlama 2.2 51s). These results prove A3’s practical decoding efficiency.
>
> ---
>
> **Q4**. Since any-order generation can, in principle, apply to structured data (e.g., image or audio tokens), do you anticipate A3 extending naturally to multimodal diffusion–AR hybrids, or would significant architectural adjustments be required?
>
> **A4**. Yes, and we believe only minimal adaptation is needed. This is due to the following points:
>
> - The formulation $P(x)=\prod_kP(x_{G_k}∣x_{G_{<k}})$ is modality-agnostic.
>
> - For image/audio tokens, the architecture remains identical; only tokenization (VQ, semantic tokens) needs to be adapted.
>
> Thus A3 can serve as a **natural AR–diffusion hybrid** for multimodal generative modeling.
>
> ---
>
> Thanks again for your detailed comments, which are very helpful, and hope our response could address your concerns. Please let us know if you have additional questions.

---

### Comment · Area_Chair_w9uN · 2025-11-28
**Please read the rebuttal and participate in the discussion**

Dear Reviewers,

Thanks for your effort in reviewing the manuscript.
Now the authors have provided the rebuttal and it's highly recommended to take a look and give your feedback.
Thanks.

AC.

---

### Author Response · Authors · 2025-12-04
**Rebuttal Summary**

Dear Program Chairs, Senior Area Chairs, Area Chairs, and Reviewers,

We sincerely appreciate the tremendous efforts of the Program Chairs, Senior Area Chairs, and especially Area Chairs in coordinating the review process. We also extend our sincere thanks to all Reviewers for their constructive and detailed reviews.

In light of the recent updates to the ICLR system and the score rollback, we provide the following summary of the discussion phase to assist the Area Chair in tracking the progress of our rebuttal. Below, we outline the main concerns raised by the reviewers and the additional experiments and clarifications we provided. The main revisions are summarized as follows:

**Additional experiments.** The primary concerns across reviewers centered on the scope of our experimental evaluation. In response, we conducted a series of additional experiments according to their suggestions and added the results in the revised version of our paper:

1. Comparison with AR variants baseline (speculative decoding), now presented in `Table 6` in `Appendix D` (Raised by Reviewers `6bME`, `yNMv` and `ky8C`).
2. Abalation study on curriculum schedule, now presented in `Table 2` in `Section 4.3` (Raised by Reviewers `6bME` and `ky8C`).
3. Practical speed–quality trade-offs compared to baselines, now presented in `Table 8` in `Appendix D` (Raised by Reviewers `6bME` and `TFMj`).
4. Results with different amount of training data, now presented in `Table 3` in `Section 4.3` (Raised by Reviewers `TFMj`, `yNMv` and `ky8C`).
5. Results on longer contexts, now presented in `Table 9` in `Apendix D` (Raised by Reviewer `TFMj` and `yNMv`).
6. Comparison with previous works ASSD [1], now presented in `Table 7` in `Appendix D` (Raised by Reviewer `yNMv`).
7. Latency number of infilling tasks in Table 1, now claimed in `Section 4.2` (Raised by Reviewer `ky8C`).
8. Resuls across different context lengths, now presented in `Table 4` in `Section 4.3` (Raised by Reviewer `ky8C`).

**Additional clarifications.** Besides experiments, we also provided additional discussions and clarifications to address the concerns of reviewers, including

1. More discussions and comparison between our proposed method and previous paradigms (AR and diffusion models), now added in `Section 2.2` and `Appendix B`(Raised by Reviewer `6bME` and `TFMj`).
2. More discussions on the distributional correctness, including comparison with previous method ASSD [1], now presented in `Appendix D` (Raised by Reviewer `TFMj` and `ky8C`).
3. More discussions on the group choice. (Raised by Reviewer `yNMv` and `ky8C`)

Overall, we believe that these additional experiments and clarifications address the concerns raised by all reviewers and further strengthen the empirical and conceptual contributions of our work.

Thank you once again for your time, effort, and dedication to the community.

Best regards,

Authors

---

### Meta-Review · Area_Chair_gSCH · 2026-01-01

**Summary:**

This paper extends the any-order permutation approach introduced in XLNet by incorporating a group-wise (any-subset) factorization, enabling parallel decoding of multiple tokens simultaneously. This modification elegantly unifies permutation-based and group-wise factorization methods while preserving the autoregressive training paradigm.

The reviewers initially raised concerns regarding the distinctions from XLNet, speculative decoding, and masked diffusion models, as well as the sufficiency of the empirical evaluation. After reviewing the revised submission, these concerns have been largely addressed through the addition of new experiments and expanded discussion.

Consequently, I have decided to accept this paper. I encourage the authors to further strengthen the camera-ready version by including more detailed comparisons with XLNet, speculative decoding, and masked diffusion models, and by carefully incorporating the reviewers’ remaining comments to further improve the overall quality of the work. Please also ensure that the newly added experiments, such as speculative decoding experiments using Llama-3.1-8B as the target model, are included in the final version.

**Reviewer Concerns:**

Addressed:

1) Distinctions from XLNet, speculative decoding, and masked diffusion models
2) Abolition study on the curriculum schedule.
3) More experiments with equal data size and equal parameter-level.

There are no significant outstanding concerns. While the issue that "parallel generation is only heuristically correct, not distributionally correct" is not fully resolved, I do not view this limitation as substantial enough to warrant rejection.

**Reviewer Scores:**

Reviewer 6bME:  The score should remain positive as 6 since the evaluation scope and ablations are expanded.
Reviewer TFMj: The score would be changed from 2 to 6 since almost all concerns are well addressed in the discussion.
Reviewer yNMv: The score would be increased to 6 due to the inclusion of new experiments and comparisons.
Reviewer ky8C: The score should be kept positive as 6 since the required new experiments are given.

---

### Decision · Program_Chairs · 2026-01-26

Accept (Poster)